# Using X-ray tomoscopy to explore the dynamics of foaming metal

Francisco García-Moreno [1,2], Paul Hans Kamm [1,2], Tillmann Robert Neu[1,2], Felix Bülk[1,2], Rajmund Mokso[3], Christian Matthias Schlepütz [4], Marco Stampanoni[4,5] & John Banhart[1,2]

The complex flow of liquid metal in evolving metallic foams is still poorly understood due to difficulties in studying hot and opaque systems. We apply X-ray tomoscopy –the continuous acquisition of tomographic (3D) images– to clarify key dynamic phenomena in liquid aluminium foam such as nucleation and growth, bubble rearrangements, liquid retraction, coalescence and the rupture of films. Each phenomenon takes place on a typical timescale which we cover by obtaining 208 full tomograms per second over a period of up to one minute. An additional data processing algorithm provides information on the 1 ms scale. Here we show that bubble coalescence is not only caused by gravity-induced drainage, as experiments under weightlessness show, and by stresses caused by foam growth, but also by local pressure peaks caused by the blowing agent. Moreover, details of foam expansion and phenomena such as rupture cascades and film thinning before rupture are quantified. These findings allow us to propose a way to obtain foams with smaller and more equally sized bubbles.

[1] Institute of Applied Materials, Helmholtz-Zentrum Berlin für Materialien und Energie, Hahn-Meitner-Platz 1, 14109 Berlin, Germany. [2] Institute of Materials Science and Technology, Technische Universität Berlin, Hardenbergstr. 36, 10623 Berlin, Germany. [3] MAX IV Laboratory, Lund University, 22100 Lund, Sweden. [4] Swiss Light Source, Paul Scherrer Institute, 5232 Villigen, Switzerland. [5] Institute for Biomedical Engineering, ETH Zürich, 8092 Zürich, Switzerland. Correspondence and requests for materials should be addressed to F.G-M. (email: garcia-moreno@helmholtz-berlin.de)

Foams are dispersions of gas bubbles in a liquid. Their arrangement is mostly disordered and bubble sizes vary. They are only temporarily stable as gravity and capillary forces displace both liquid and bubbles. Foams of almost any possible liquid exist and both liquid and solidified foams find a wide range of applications: aqueous foam (shaving cream, champagne, foam in fire extinguisher), polymer foam (packaging material), food foam (chocolate), ceramic foam (filters), glass foam (insulation material) and, finally, metal foam[1]. The latter has been known for a long time[2] but its application in light-weight construction, crash energy absorption or other purposes is still developing[3,4].

Liquid foams are complex in terms of structure, properties and temporal evolution. Their structural elements are films that are connected by so-called Plateau borders that meet in nodes[5]. Films are prone to rupture, and rupture induces bubble coalescence. Liquid metal foams are difficult to study owing to high temperatures and their opacity to light, which is why they are much less explored than, for example, aqueous foams. Similarities are expected but how exactly problems such as coarse or non-equally sized bubbles can be solved is not known.

To generate foams, gas has to be added to a liquid, for example, by shaking, sparging gas into it, by gas dissolution and subsequent pressure release, or by decomposition of blowing agents. For making metal foam, one particular method is especially efficient: metal and blowing agent powders are mixed, compacted to an almost dense foamable precursor and then heated to the melting temperature of the alloy, during which the metals softens, the blowing agent releases gas and bubble nucleation and growth set in. The foam expands whereas the gas bubbles grow and displace the liquid. By freezing the foam in different expansion stages one can map foam evolution and study its structure[6–8] but such studies do not reveal much of the physics governing foaming, which is why in-situ studies have been initiated, in the easiest case by measuring the kinetics of foam volume change[9].

During heating of the precursor material sub-μm sized bubble nuclei are created. They have so far been resolved only by indirect methods such as ultra small-angle neutron scattering[10] or by scanning electron microscopy of sectioned samples[10,11]. The nuclei then grow into bubbles large enough to be investigated ex situ by optical microscopy or in situ by X-ray radioscopy. The latter yields projection images of features along a given viewing direction and also allows one to distinguish bubbles and to detect defects[12,13] but with the serious limitation that non-connected features may appear superimposed and that small (early) bubbles remain invisible. Switching from (2D) radioscopy to (3D) tomography lifts these limitations, but until recently tomography has been too slow to cope with the constant movement of expanding foams. The various processes observed in a foam take place on characteristic time scales: for bubble nucleation and growth typically on the scale of 100 ms to 1 s. Topological transitions and the thinning of films are faster, 5 ms to 100 ms. The time for film rupture is then well below 1 ms[14]. These time scales define the necessary time resolution of an appropriate imaging method, see Fig. 1, and make clear that new methodological developments are required.

Here, we give a detailed description of liquid metal foam evolution and elucidate key phenomena such as bubble rearrangements, bubble collapse and film rupture by applying real-time in situ X-ray tomography at acquisition rates up to 208 tomograms per second (abbreviated tps) continuously over the entire foaming process. We further stretch the time resolving power of tomography by modifying the reconstruction techniques and do all this at voxel sizes of 4.9 μm and spatial resolutions of ≤15 μm while maintaining a reasonable field of view. The work is enabled by our recent experimental improvements that speed up

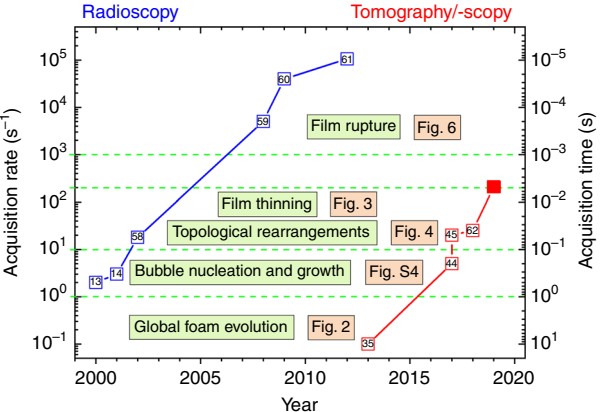

**Fig. 1** Image acquisition rates and times used to study metal foams. The highest image acquisition rates in each year achieved for 2D radioscopic (blue, in s$^{-1}$, fps)[13,14,58–61] and 3D tomoscopic (red, in s$^{-1}$, tps)[35,44,45,62] analyses of evolving liquid metal foams are given. The horizontal broken lines delimit the image acquisition rate ranges required for the study of the phenomena named in the boxes highlighted in green. Figure numbers refer to figures of this paper. The numbers in the small open squares are references. The full red square marks the current work

tomography so much, that 3D images can be obtained in a movie-like mode with now ≥200 tps, 10 times more than possible previously. The key to this success has been the construction of a new sample rotation stage, the ability to transform the huge amount of data from the camera chip to storage systems continuously for minutes and new automatised data processing procedures.

## Results

**Tomoscopy**. We shall use the word tomoscopy for time-resolved studies explicitly aiming at resolving the dynamics of materials continuously in 3D. Therefore, in the same way a temporal dimension is added to radiography to form radioscopy, tomography evolves to tomoscopy. We avoid using attributes such as ultra-fast, which appear in the literature for much slower time-dependent tomography (0.1 tps[15] vs. 208 tps here).

**Foam evolution**. Figure 2c shows the number of bubbles (black) in and the total sample volume calculated from individual tomograms (blue) of an AlSi8Mg4 precursor containing 0.25 wt.% TiH$_2$ as it converts into a foam during melting and release of hydrogen caused by heating to 680 °C. One tomogram per second (1 tps) was recorded continuously from the beginning to after the end of the entire foaming process marked by the onset of solidification after stopping to heat. In total, 300 individual tomograms were reconstructed and analysed by automatised tools.

Both volume expansion and the absolute number of bubbles clearly evolve in distinct stages: stage I up to 87 s, with temperatures ranging from ~450 °C to 560 °C, features many newly appearing bubbles, but the overall volume shows only small changes. Stage II takes place between 560 °C and 590 °C. The number of pores increases by a factor of 4 and the foam notably expands before reaching an inflection point after 107 s. Stage III is observed above 590 °C (after 107 s) and the number of bubbles decreases while volume expansion shows a major boost before it finally slows down after ~175 s. The final stage IV features volume shrinkage corresponding to dropping temperatures and ensuing solidification owing to the end of heating.

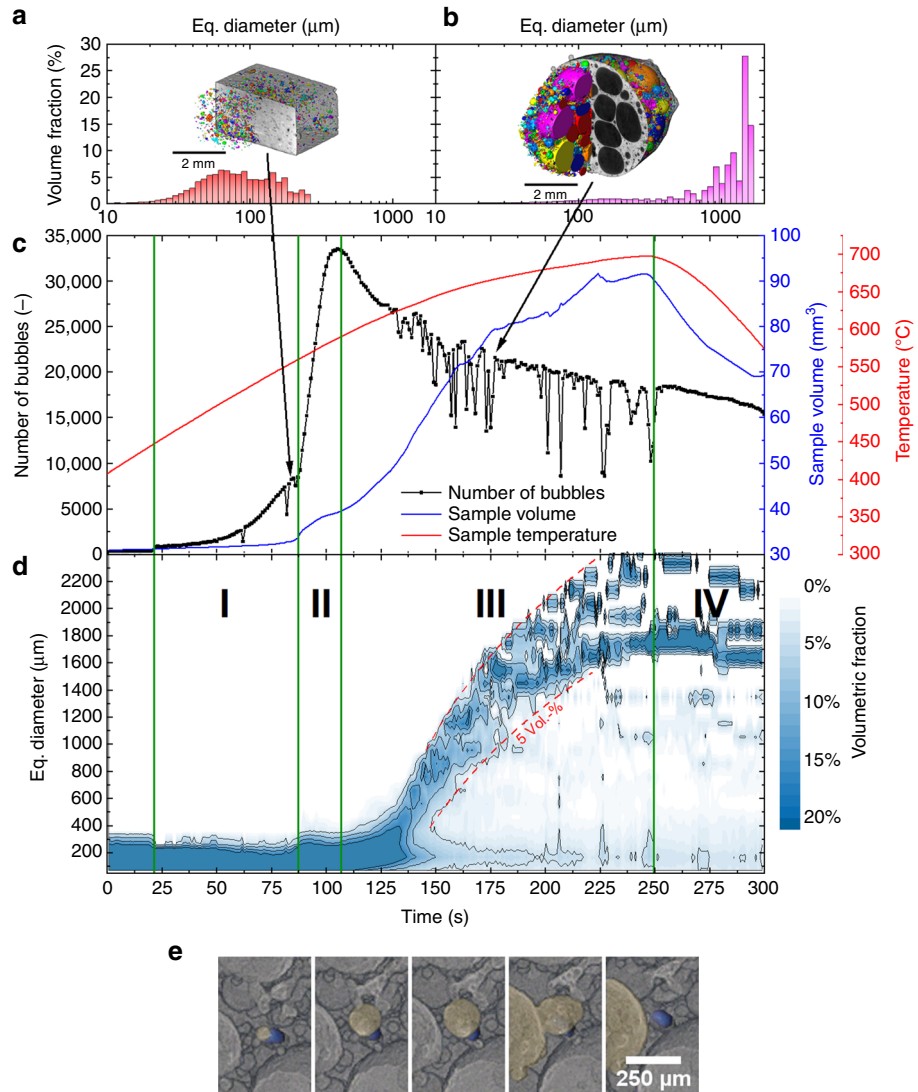

**Fig. 2** Foam evolution explored by tomoscopy. Foaming of AlSi8Mg4 alloy precursor containing 0.25 wt.% $TiH_2$ observed in situ by tomoscopy at an acquisition rate of one tomogram per second (1 tps). **a**, **b** Two segmented tomograms extracted from the series of 300 ones showing the liquid foam structure and corresponding equivalent diameter distributions at different times **a** $t = 85$ s and **b** $t = 177$ s. The different colours indicate separated bubbles. **c** Quantitative analysis of 3D data yielding the number of bubbles (black) in and the volume (blue) of the expanding foam as the temperature (red) is increased and finally decreased. Four expansion stages are delimited: I: bubble nucleation driven by adsorbed gas, II: bubble nucleation driven by decomposition of $TiH_2$, III: stage of pronounced bubble inflation and coalescence, IV: solidification and shrinkage. **d** Evolution of the bubble equivalent diameter distribution over the whole foaming process expressed by the blue scale coded volume fraction of each diameter class. Broken red lines show that the bubbles grow and the size distribution broadens in stage III. **e** Creation, inflation, and merger of a bubble (shown in yellow) into another one near a $TiH_2$ particle (shown in blue). Time increment is 1 s

Stage I can be traced back to the Al–Mg powder particles contained in the precursor that melt already at ~450 °C. The sources of blowing gas at this temperature are the surfaces of all former powder particles that release mainly adsorbed hydrogen during heating. We call this an intrinsic gas source[16]. The main nucleation stage II coincides with a rapid increase of liquid phase above the solidification temperature of AlSi8Mg4 (558 °C)[17] and an increase of the amount of hydrogen gas released by the $TiH_2$ particles[18] that rapidly augment the number of (still small) pores. Beyond ~107 s, in stage III, many bubbles merge to larger ones (coalescence) and the number of bubbles decreases. The strong volume expansion in this stage is caused by a pronounced inflation of individual bubbles.

The bubble count in Fig. 2c shows pronounced dips at some times. These are motion artefacts during the 1 s acquisition period

of an individual tomogram and are caused either by movements of the entire sample (e.g., at 60 s) or by the rupture of films in the expanded foam (after 130 s). Supplementary Fig. 3 shows an example for a distorted tomogram occurring at a foaming time of 207 s. A faster tomography acquisition rate avoids this problem, see Supplementary Fig. 4, but then the sample has to be reduced in size and will grow to beyond the field of view owing to which counting bubbles is affected.

Figure 2d demonstrates that bubbles not only grow tremendously in size during expansion but also that their distribution broadens, showing that bubble coalescence is not a statistically equally distributed phenomenon but is very strong in certain areas where very large bubbles are formed. One key point of metal foam development is to know the mechanisms of coalescence, which factors lead to it and how one might be able to reduce it.

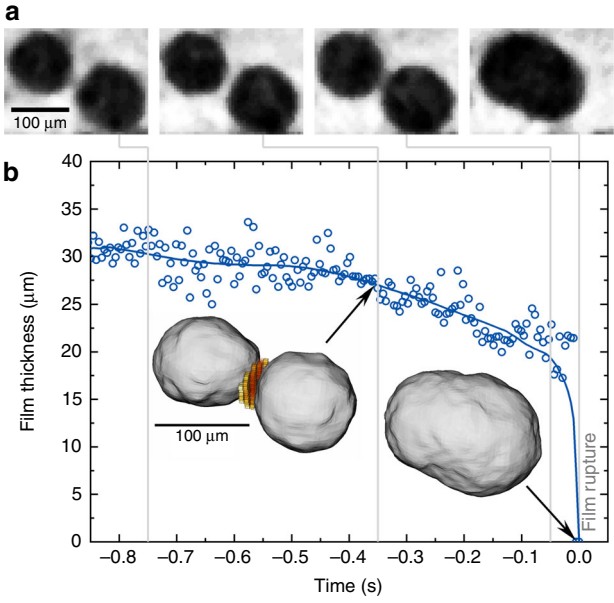

**Fig. 3** Phenomena preceding film rupture as measured by X-ray tomoscopy. Liquid foam (same alloy as in Fig. 2) is observed by tomoscopy with 208 tps acquisition rate. **a** Four tomographic slices at different times (grey lines) showing two approaching bubbles. **b** Film thickness over time shown for a period of ~1 s before the film between the two bubbles ruptures at $t = 0$ s. Insets: rendering of the two bubbles and the corresponding film separating them (given in red and yellow) and the resulting bubble after coalescence

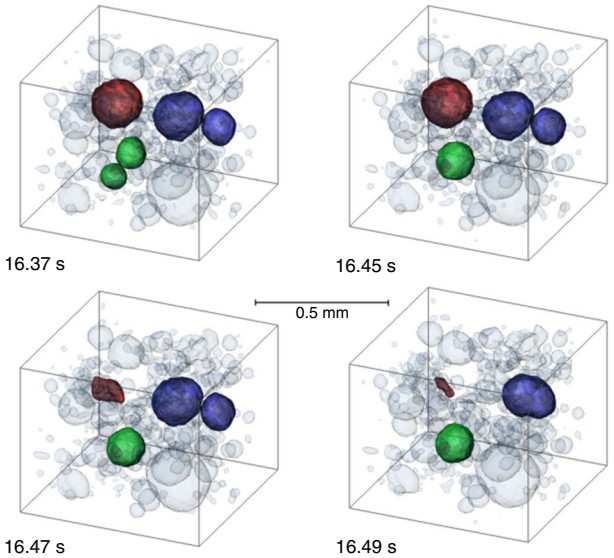

**Fig. 4** Evolution of a group of five bubbles in a liquid metal foam. 3D rendering of topological rearrangement of bubbles in a close neighbourhood inside a volume of $0.74 \times 0.74 \times 0.63$ mm³. A cascade of events can be seen starting with the merger of two bubbles (green), collapse of another (red), and another merger of two bubbles (purple). Same alloy and tomoscopy rate as in Fig. 3

The reason for coalescence is not yet known other than in aqueous foams, where owing to capillary forces and gravity induces drainage, i.e., flow of liquid out of the films, films become thinner and eventually the film-stabilising disjoining forces stemming from electrostatic forces of the polar surfactant molecules are overcome[19–22]. The actual rupture process has been observed using fast optical filming and reveals an instability occurring in the middle of a circular film and subsequent rapid spreading[23–25].

**Film rupture**. Single metal films have been investigated both optically and by X-ray radioscopy[26]. The mechanism of rupture was found to differ from that in aqueous foams as the oxide layers residing on both sides of each metal film play a role. They sandwich the liquid metal and contribute to film stabilisation[27]. Shortly before rupture sets in, small translucent areas appear, indicating the retraction of melt from the thinnest part of the film[26]. This picture can be proven using tomoscopy at an acquisition rate of 208 tps to measure the thickness of a film separating two bubbles. In Fig. 3 we find that within the last second before rupture (defined as $t = 0$ s) melt flows out of the area between the two bubbles as evidenced by a decreasing film thickness and that this flow accelerates the closer to rupture the film gets. The minimal (final) thickness of the film is ~20 μm. Thus, we are able to show that metallic films break after too much liquid has been removed. The rupture must then have been triggered by some (not yet resolved) incident to be discussed below.

**Bubble coalescence**. Figure 4 shows four images taken from a tomoscopy series also acquired at 208 tps. There is a group of bubbles in the small sub-volume depicted that undergoes changes in neighbourhood and size within 120 ms. First, the two bubbles in green coalesce to a larger one. Then, the bubble shown in red

starts collapsing rapidly within 40 ms (this collapse is further detailed in Supplementary Fig. 5). Finally, the two bubbles in purple suffer coalescence. Thus, it is clear that collapse or rearrangement of bubbles can trigger further such events. The rearrangement of liquid caused by coalescence disturbs the local system and forces must be acting on neighbouring bubbles that lead to a cascade of such events similar to what is known for aqueous foams[28–30]. Radioscopy on metal foams has also shown such correlated events but owing to the superposition of features along the viewing direction it is not always clear whether temporally correlated ruptures belong to the same cascade[31].

## Discussion

Two mechanisms have been discussed as possible triggers for bubble coalescence: first, for metal foams blown with an agent such as $TiH_2$, expansion coalescence induced by growth has been postulated[7] as during foam expansion films might eventually be stretched to beyond the limit of stability. Moreover, it has been suspected that drainage induces drainage coalescence, as it removes liquid from aqueous[22] and metallic films[32].

Undoubtedly, the first mechanism is acting in the stage of rapid foam expansion but we observe coalescence also at a time foam expansion has already levelled off. Tomoscopy yields evidence that coalescence is also pronounced near a strong local gas source as seen in Fig. 2e, where a $TiH_2$ particle (blue) is visualised. Right next to this particle fast bubble growth followed by a coalescence event is observed. Other such events can be seen in the corresponding video (Supplementary Video 2). A more direct proof for this trigger mechanism of coalescence is obtained by comparing samples that are foamed with the blowing agent $TiH_2$ and others that owe their expansion exclusively to the gas released by the Al-Mg particles (intrinsic gas source)[16] that causes stage I in Fig. 2. To obtain a comparable volume expansion, a pressure drop during foaming is applied[33]. Film rupture events are counted by analysing images obtained by X-ray radioscopy (tomoscopy is neither necessary nor possible in this case). Figure 5a shows that the cumulated number of rupture events is around one order of

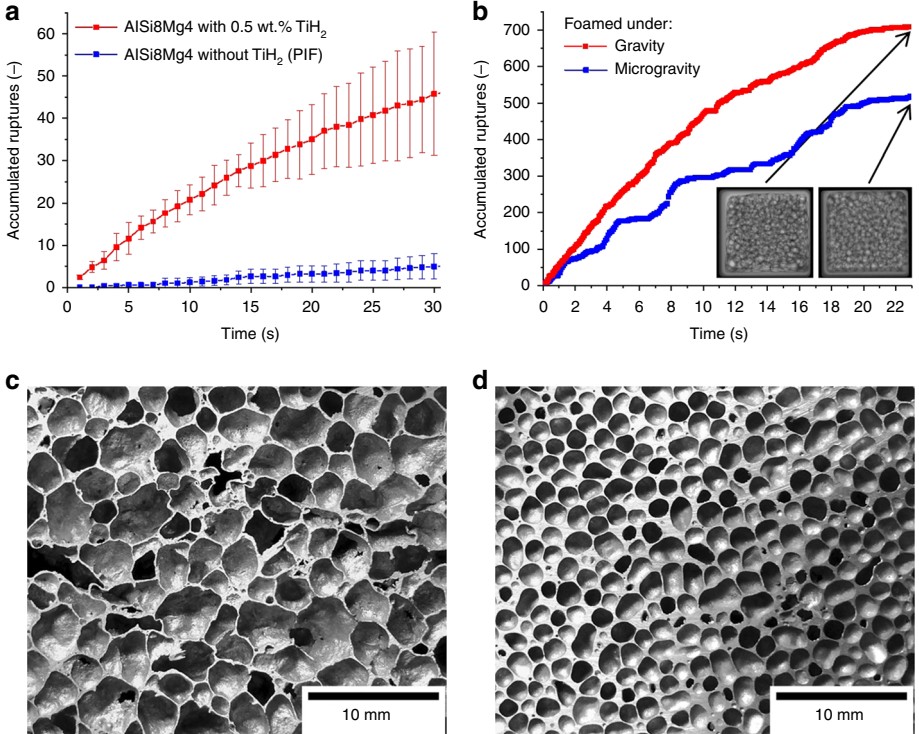

**Fig. 5** Experiments proving that coalescence is caused both by the blowing agent and gravity. **a** Cumulated number of film rupture events in AlSi8Mg4 foams determined by evaluating X-ray radioscopies over a period of 30 s. Samples of similar porosity foamed with TiH$_2$ are compared with samples foamed without. In the latter case, volume expansion was supported by a pressure drop imposed on the sample (pressure-induced foaming—PIF)[33]. Five experiments were averaged, error bars correspond to the standard deviation. **b** Film rupture events in AlCu10Mg15 foam determined by analysing X-ray radioscopies obtained under normal and microgravity conditions, both at ambient pressure. Two images are shown for the final state marked by arrows. **c** Cross-section of an AlSi8Mg4 foam blown with TiH$_2$ compared with **d** AlCu10Mg15 foam produced without TiH$_2$ featuring a more homogeneous distribution of smaller pores

magnitude higher in the foams containing TiH$_2$ than in the ones made without. TiH$_2$ gives rise to locally high gas pressures, whereas in the TiH$_2$-free precursors the gas filling the bubbles is released from the surfaces of the powder particles and as pressures are small and the particles uniformly distributed, films are less prone to rupture. We can therefore state that local growth coalescence in addition to global growth coalescence constitutes another mechanism.

To assess the further mechanism (gravity-induced drainage coalescence) we carried out two foaming experiments under equal conditions, one under normal gravity, the other under microgravity. As it was not possible to apply pressure variations to the experimental setup under microgravity, a different alloy (AlCu10Mg15) was chosen that can be foamed to considerable volume without using a blowing agent just relying on the gas released from the Al–Mg powder particles within the precursor. Figure 5b shows that under conditions of microgravity there are ~40% less coalescence events after 22 s of foaming. Thus, removal of liquid from a film can also trigger coalescence but to less extent than the pressure differences induced by the blowing agent. Thus, drainage coalescence is much weaker than both global and local growth coalescence. Figure 5c, d compares solidified foams produced with TiH$_2$ and without. The latter foam exhibits smaller pores that are more evenly distributed, which is the consequence of reduced coalescence.

The rupture of films in metal foams is actually not resolved in time as it takes place in <1 ms, i.e., below the time resolution of our tomoscopy (4.8 ms) as we know from radioscopy studies[13]. Although 208 tps are nominally not sufficient, we can contribute

to understanding such fast phenomena by using an incremental reconstruction technique further explained and visualised in Supplementary Fig. 2. Figure 6 shows the merger of four bubbles into one. The bubbles in question are given in Fig. 6a, b as two (normal fixed-block) tomograms featuring four bubbles collapsing into a larger one in less than the time between the tomograms. From another event shown in Fig. 4 we deduce that the collapse of the four bubbles must be mutually induced as a cascade and not simultaneous, just too fast to be resolved. Figure 6e then demonstrates for five different pairings of bubbles (colour-coded) a stage of clearly separated bubbles (Fig. 6e, top row) and blurred contours of bubbles indicating that the rupture event has taken place during acquisition of the 192 projections the slice is based on (2nd and 3rd rows). The 4th row in all but the middle column shows the bubbles shortly before completion of merger, whereas in the last row the coalesced bubble is seen in its final state. The sequence of bubble coalescences is best seen in Fig. 6d, where bubbles 1 and 2 start merging shortly before they join bubble 3 and bubble 4 is absorbed last, showing a cascade rupture behaviour.

We use Fig. 6e to estimate the time it takes to complete the rupture of a film from its onset after a hole or a crack has formed in the film until the remnants of the film have been absorbed into the surrounding liquid (but before the merged bubble has reached a spherical shape). This can be illustrated looking at the 2nd and 3rd rows of the 5th column (cyan coloured box) in Fig. 6e. We notice an onset of rupture in #56, whereas in #84 there are still traces of the former film connecting the bubbles. The elapsed time is $28 \times 25\,\mu s = 700\,\mu s$. In the 4th column, however, image

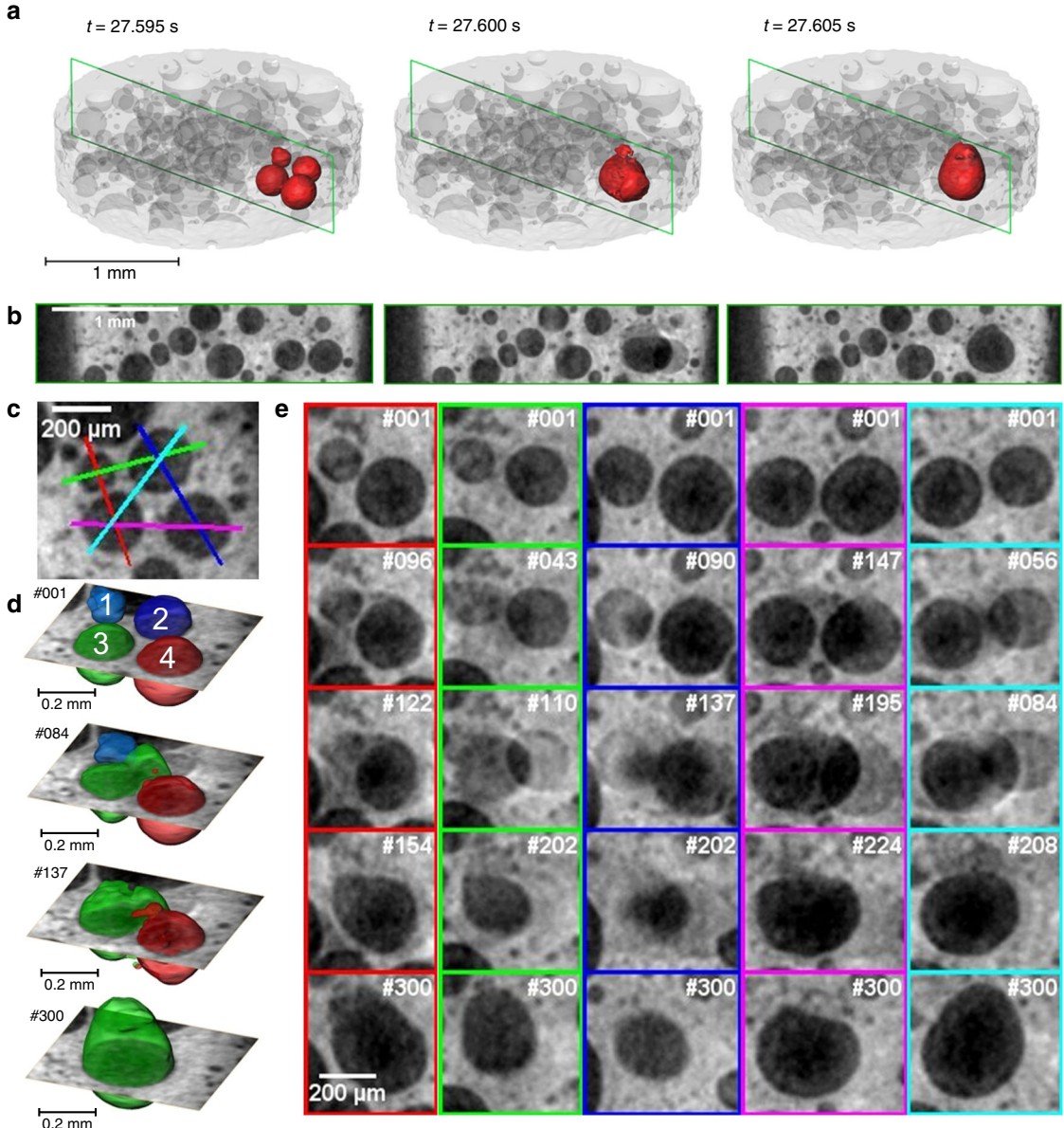

**Fig. 6** Analysis of the coalescence of bubbles by tomoscopy at 208 tps (AlSi8Mg4 alloy). **a** 3D rendering of structure highlighting the bubbles under investigation. **b** Corresponding perpendicular slices of the structure. **c** Definition of five colour-coded planes through the four bubbles in **a**. **d**, **e** Images/slices obtained by the incremental reconstruction technique described in Supplementary Fig. 2. Numbers #nnn refer to time increments of 25 μs starting from an arbitrary point. **d** 3D rendering at various times showing the original configuration of numbered bubbles (#001). The configuration changes after 2.08 ms in #084 where bubble 2 and 3 have merged to a dumbbell-shaped intermediary bubble. In #137, (3.83 ms from the beginning) bubble 1 has completely merged into the dumbbell-shaped one. Even later, after 5.6 ms (#300) bubble 4 has also joined. **e** Slices in the planes defined in **c** showing the merger of bubbles in **d** in more detail

#147 shows onset of coalescence, whereas in #195 the surfaces of the newly formed bubble are smooth. The time elapsed is 48 × 25 μs = 1200 μs in this case. Thus, it can be estimated that the rupture time is around or slightly below 1 ms. This value is higher than values determined by radioscopy in similar foams[13], probably caused by the overlap of frames. Still, incremental reconstruction allows to distinguish events taking place about five times faster than the separation of tomograms.

In conclusion, new insights into fast phenomena in liquid metal foam evolution have been obtained by applying the newly developed measurement technique 'X-ray tomoscopy', which currently provides up to 208 full 3D data sets of evolving samples per second at 4.9 μm voxel size over minutes. On the timescale of

seconds the nucleation and growth of bubbles can be explored, whereas the now available acquisition rates enable us to also unravel topological transitions, film thinning and bubble merger on an intermediate timescale (5−100 ms). Film rupture takes place on the shortest timescale and can be approached by applying incremental reconstruction techniques.

The combination of tomoscopy with radiography experiments under varying gravity conditions applied here shows explicitly that non-uniformities of metal foams are caused by the deleterious growth of very large bubbles in the vicinity of blowing agent particles. This suggests a way to improve metal foams. First trials have shown that this finding can be used to improve the manufacturing procedures. Figure 5c, d comparing a foam produced

using the blowing agent $TiH_2$ and another one based on an intrinsic gas source shows a possible direction. Commercial application of metal foam would greatly benefit from an upscaling of such new foaming techniques.

## Methods

**Sample preparation.** Foamable precursors of AlSi8Mg4 (in wt.%) alloy were produced by mixing elemental Al and Si powders and AlMg50 alloy powder with the blowing agent (0.25 wt.% $TiH_2$) for 20 min. The powder blend was placed in a cylindrical steel die of 36 mm diameter and cold compacted for 5 s followed by uniaxial hot-compaction at 300 MPa and 400 °C for further 15 min in the same die[17]. The almost fully dense compacted tablets were then cut into pieces and milled to samples of various sizes:

- Cylinders of 2 mm diameter and ~5 mm length to fit into the small boron nitride crucible used for tomoscopy at high acquisition rates (≥20 tps);
- 2 mm × 2 mm × 4 mm cuboids to fit into a boron nitride crucible of 8 mm diameter for experiments with an acquisition rate of 1 tps,
- 20 mm × 10 mm × 5 mm cuboids for the radioscopy experiments in normal and microgravity environments in an also cuboid crucible.

AlCuMg alloy samples for the microgravity experiments were prepared in analogy from Al, Cu and AlMg50 powders.

**Laser heating and foaming.** The tomoscopy setup (see below) contained a cylindrical boron nitride crucible with a piece of foamable material inside. The samples were heated using near-infra-red lasers, wavelength 980 nm, of 150 W power[34]. The light of two such lasers coming from nearly opposite directions in a plane perpendicular to the X-ray beam was directed onto the surface of the boron nitride crucible and formed a spot of ~4 mm × 6 mm size. The part of the laser light not intercepted by the crucible was dumped into two water-cooled beam stops behind the sample. The temperature of the sample was controlled through adjustment of the laser power and measured via a calibrated pyrometer focused onto the outer wall of the boron nitride crucible (emissivity = 0.85) at the height of the X-ray beam (see Supplementary Fig. 1). The actual sample temperature could not be measured directly, but was calibrated against the power and pyrometer readings by melting a 99.7% pure aluminium reference sample. The heating and cooling temperature profiles were defined by a simple current ramp: The laser power was first set to a value produced by a current of 10 A, then ramped to 16.5 A with a ramp time of 162.5 s, where it was held constant for a holding time of 60 s. Thereafter, the laser was turned off and the sample cooled down naturally.

**Fast and high-resolution tomography.** Time-resolved tomography for in situ analyses in materials science has become very popular[35–40]. Systems such as rechargeable batteries[41], solidifying alloys[15,34,42,43] or evolving metallic foams[35,44–46] are of special interest. Time-resolved radioscopy for very short exposure times (down to 100 ps per image) and 1 Mfps repetition rate is possible[47], but acquisition is limited to very short periods, thus not allowing to follow longer processes continuously. Tomography naturally has been much slower than radioscopy but some proofs on solid samples have shown that acquisition rates of 100 tps are feasible[48]. The state of the art in fast and high-resolution tomographic microscopy (we consider here voxel sizes smaller than 10 μm) applied to dynamically evolving materials is 20−25 tps[40,45,49]. This work is showing an increase of this parameter to 208 tps while maintaining other features such as field of view, spatial resolution and period of continuous acquisition.

**Tomoscopy setup.** Synchrotron X-ray tomoscopic experiments were carried out at the TOMCAT beamline of the Swiss Light Source, Paul Scherrer Institute, Switzerland. The polychromatic beam generated by a 2.9 T superconducting bending magnet was cut down in size to the required field of view at the sample position with the beamline front end slits and energy was filtered using 5 mm of glassy carbon and a 325-μm thick single crystalline Si wafer to suppress lower energies and limit the heat load on the sample and detector system while maintaining a high photon flux at higher X-ray energies. The edge-enhanced X-ray transmission image produced by the sample was converted to visible light using a 150-μm thick LuAG: Ce scintillator (Crytur, Czech Republic) at a distance of 260 mm from the sample and magnified by a high numerical aperture microscope (Elya Solutions, Czech Republic) with a variable magnification ranging from ~×2 to ×4. This visible light image was then recorded by the GigaFRoST high-speed camera with a pixel size of 11 μm, resulting in an effective pixel size of 4.9 μm[50]. The spatial resolution under extrem experimental conditions (at the edge of the sample for a tomogram recorded at 200 tps) was measured to be ≤15 μm.

The new, self-developed high-speed rotation stage allowed for an operation at frequencies $f > 100\ s^{-1}$ while maintaining a very precise alignment of the rotation axis and ensuring accurate time-to-angle stability (Supplementary Fig. 1). With this, >200 tps can be acquired (one half-rotation required for each tomogram). Supplementary Fig. 1b shows the optical microscope with the protected scintillator screen at the bottom and the GigaFRoST CMOS camera at the top. Supplementary Table 1 lists the data acquisition settings used for the experiments. Special attention

has to be paid to centring the samples or crucibles and to maintain a precise rotation axis without tumbling errors (avoiding eccentricity and wobble). The hitherto used stepper motor setups proved to be unsatisfactory. This is why a new rotation stage was constructed. It consists mainly of an inrunner brushless servo motor with Hall sensors and a position encoder. The crucible is placed directly on the motor shaft. The associated positioning and velocity control unit ensures a very precise angular velocity accurate enough for our purposes.

Another effect to consider are centrifugal forces. The radial acceleration is $a = 4\pi^2 f^2 r$. In our case, 208 tps correspond to $a = 42\ m\ s^{-2} \approx 43\ g$ at the sample cylinder walls ($r = 1$ mm, $f = 104$ Hz). It is very likely that foams in late stages of expansion will be affected by such forces. In the present study, however, we investigate small foam samples in early stages of expansion that contain much smaller bubbles than fully expanded foams. The compressibility of such foams is much lower than of expanded foams and the alloy AlSi8Mg4 used is known to be less prone to drainage than others[17]. The small influence of gravity is confirmed by visual inspection of the foam through the radiographic preview monitor that does not show effects of drainage, i.e., bubbles floating to the rotation axis as rotation speed is increased. In order to verify this impression, radial density profiles of foams investigated by tomoscopy at 1 tps (max. 0.001 g radial acceleration), 50 tps (max. 2.5 g) and 208 tps (max. 43 g) were calculated in a region of interest to avoid edge effects and are shown in Supplementary Fig. 6. For that purpose the intensities of the inverted grey images for early (1% porosity) and later foaming stages (40% porosity) were averaged over the number of slices and normalised to the arithmetic mean of the respective intensities of the (nearly dense) original sample precursor with a porosity of 1%, whereupon they were represented over the radius normalised to the circumference. There we can observe that, apart from the natural local density scattering of metal foams, there is no notable difference between the radial density profiles.

**Data acquisition.** Owing to the fast dynamics and the long duration of the foaming processes, the demands on data acquisition are challenging. Measurements at 40 kHz frame rate need to be sustained over several tens of seconds (in some cases up to 2 min), thus producing millions of images. Paul Scherrer Institute (PSI) has developed the GigaFRoST camera and readout system precisely for this task[50]. In short, the data from a high-speed imaging chip is streamed directly to a backend server via eight parallel fibre-optical data connections, yielding transfer rates of ~8 GB $s^{-1}$. The backend server is equipped with 512 GB of memory and can thus sustain these data acquisition rates for ~60 s for 528 × 128 pixels image size. From the start of acquisition, data are immediately written to a high-performance file system. In addition, a subset of the acquired projection images is sent to a preview system at the beamline with a configurable frame rate to visually monitor the evolution of the sample. The adjustable frame rate allows one to always display the identical viewing direction through the sample. This selected angular view acts as simulated radioscopy and is an important tool for controlling the process during the experiment and allows for reacting to incidences or to decide whether an experiment can be stopped.

**Data reconstruction.** Tomographic reconstruction of the time-dependent volume data was performed on blocks of $N$ images extracted from the whole data set. The raw projection data were first filtered using the propagation-based phase contrast algorithm by Paganin et al.[51] with a δ/β ratio of 700 to enhance the projection contrast, and then reconstructed with the ASTRA-Toolbox[52] or with the *gridrec* algorithm[53].

An incremental reconstruction (IR) approach based on advancing the 'reconstruction window' (a so-called 'sliding window') stepwise through the acquired projections was applied in cases where a higher time selectivity was required (e.g., Figure 6). For conventional reconstructions such as in Figs. 3 and 4, individual tomograms are based on 192 projections taken over a 180° rotation of the sample, where projections are taken every 25 μs and the entire tomogram thus takes 192 × 25 μs = 4.8 ms. Subsequent tomograms are based on fixed and disjoint blocks of 192 projection images. For incremental reconstruction as used for Fig. 6, however, reconstructions are taken for blocks of radiographic projections that may overlap as schematically explained in Supplementary Fig. 2. The offset between two blocks of images is 25 μs.

A sudden event such as film rupture might happen at the transition from one fixed-block tomogram to another but will eventually move into the incrementally reconstructed tomograms in between. If the sudden event is spread out over a small period of time the IR tomograms will show its gradual appearance and allow for an assessment of the time the event takes.

**Data analysis.** The large volumetric and time-resolved data sets require not only an extended data handling infrastructure and batch reconstruction options but also a batch analysis mode, which was not available at any beamline and it seemed unlikely that beamlines would be able to develop those just for special applications such as metal foams as the definition of an analysis workflow only makes sense in the context of a particular experiment. Owing to the great variety of sample types there is no single solution to the analysis problem that will solve all tasks. Moreover, the type of information required varies between experiments and users. Therefore, we developed our own workflow using MATLAB to quantify in a batch

modus several parameters such as number of pores, total porosity, equivalent diameter of pores, etc. as e.g., shown in Fig. 2.

**Motion artefacts**. Motion artefacts occur whenever the sample evolves 'too much' during acquisition of radiographic projections over 180°, in the sense of being shorter in time or locally larger than the spatio-temporal resolution of tomoscopy. This can happen when sudden motions occur in the liquid foams under study. Supplementary Fig. 3 shows an example of a film rupture that adds blurriness to the corresponding images.

An estimate for the velocity $v_{evol}$ of a feature in an image for a given pixel size $x$ and time for one tomography $t_{tomo}$ is given by

$$\nu_{evol} \leq \frac{x}{t_{tomo}} \tag{1}$$

where $x = 4.9$ μm and $t_{tomo} = 4.8$ ms (for 208 tps acquisition time, see Supplementary Table 1). We obtain a critical velocity of $v_{evol} = \sim 1$ mm s$^{-1}$, above which blurring should be detectable. For 1 tps as applied in Supplementary Fig. 3, $v_{evol} = \sim 5$ μm s$^{-1}$. This criterion is very strict (one pixel blurring) and blurring might not inhibit analysis even if $v_{evol}$ is exceeded. In Supplementary Fig. 3, however, blurring has a serious impact on the tomogram.

We carried out a similar experiment as in Fig. 2 based on 20 tps acquisition rate, see Supplementary Fig. 4. In this case, the field of view becomes too small to capture the entire foam in advanced foaming stages. This is why the bubble number (expressed as a density here) drops much more than in Fig. 2 after the maximum pore number has been reached. Moreover, the temperature course is slightly different. Beside these restrictions, the general course of bubble number is comparable just that there are no motion artefacts and that features such as the transition from slow to fast expansion at around $t = 70$ s is much better resolved in time.

**Microgravity experiments**. Microgravity experiments were performed on the 65th and 67th parabolic flight campaign of the European Space Agency on board of an Airbus aeroplane[54]. The radioscopic setup contains an exchangeable furnace, an X-ray diagnostic system, and a gas cooling system. Aluminium alloy samples were melted and foamed inside a crucible of 4 cm$^3$ volume. The resistive furnace can be controlled accurately (temperature accuracy $\pm 1$ K) during heating and foaming. The X-ray diagnostic system consists of a micro-focus X-ray source, a high-resolution digital CMOS X-ray detector and an image acquisition computer. Images acquired during the microgravity phase ($\sim 22$ s) were stored on a solid state disk on board and copied for further analysis on ground after the flight. The quantification of film ruptures from the image sequences was done automatically by a self-developed software. More details are described elsewhere[55].

**Radioscopic experiments**. For the comparison of foaming and film stability in samples foamed by the blowing agent TiH$_2$ and others not containing TiH$_2$, complementary radioscopic experiments were performed in a laboratory-based X-ray micro-focus imaging diagnostic system. The same alloy was foamed with TiH$_2$ at ambient pressure or without TiH$_2$ by pressure-induced foaming under otherwise similar conditions[33]. Quantification of film rupture in liquid metal foams was done by image analysis as described elsewhere[55].

**Further example**. A further phenomenon observed is that of bubble collapse. This must not be confounded with the well-known coarsening phenomenon in foams caused by diffusion of gas from smaller bubbles to larger ones, as this effect is very slow in metals[56,57]. In Supplementary Fig. 5a, gas must be escaping from the interior of the bubble to the exterior of the sample through an open channel, which cannot be seen directly, but this view point is supported by the proximity of the bubble to the surface (200 μm). The kinetics of this gas diffusion is quantified in Supplementary Fig. 5b, where the pore volume is visualised and given in time steps of 5 ms. The volume decreases from an initial value to 1/40 of that value within $\sim 30$ ms. The rate of gas loss is initially the highest and then levels off. This shows the influence of the rigid solid oxide layers on the inner wall of the bubble as a completely elastic bubble surface would shrink at an increasing rate as pressure scales with the inverse of bubble diameter according to Laplace's law. These oxides also lead to the wrinkly irregular shape of the collapsed bubble.

## Data availability
The experimental data and the workflow using MATLAB to quantify in a batch modus are available from F.G.M. upon reasonable request.

## Code availability
The processing reconstruction algorithms (propagation-based phase contrast algorithm by Paganin et al.[51], ASTRA-Toolbox[52] and gridrec algorithm[53]) are open source and available in the internet.

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

## Acknowledgements

The European Space Agency provided funding through MAP AO99-075 and arranged the 65th and 67th parabolic flight campaigns. The Deutsche Forschungsgemeinschaft funded the work through Ga 1304/5-1 and Reinhart-Koselleck project Ba 1170/40 and the German Bundesministerium für Bildung und Forschung through project 05K18KTA. We acknowledge the Paul Scherrer Institut, Villigen, Switzerland for providing synchrotron radiation beam time at the TOMCAT beamline X02DA of the SLS. We thank Nadine von der Eltz for preparing samples.

## Author contributions

F.G.M., P.H.K. and T.R.N. contributed equally to this work in designing the experimental setup and developing measurement procedures. M.S., C.M.S. and R.M. developed the beamline setup and characterisation method. F.G.M, P.H.K, T.R.N, F.B. and C.M.S carried out the various experiments. Data analysis was mainly done by P.H.K and R.M., J.B. and F.G.M. wrote the manuscript with contributions from all the authors.

## Additional information

**Competing interests:** The authors declare no competing interests.

