## [Transparent Peer Review File · Nature Communications]

Reviewers' comments:

Reviewer #1 (Remarks to the Author):

Summary

The authors have demonstrated capability to obtain 3-d (tomographic) X-ray images at an order of magnitude higher rate than reported previously. They have applied this technique to observe the structure changes in a metallic foam produced by heating an alloy containing outgassing agents. The technique being performed at a synchrotron (where the X-ray beam is necessarily fixed) requires that the sample be rotated at 100rpm. The influence of the inertial forces pertaining on the sample have been compared by contrasting with radiography results obtained in microgravity. They have shown dramatic new capability in X-ray imaging, and applied it to a problem of both technological importance and fundamental scientific interest. The paper should be published and I think it is suitable for Nature Communications.

One thing that was not clear to me, is what is the main reason or key technology allowing the greater speed, is there greater flux at Tomcat than previously? or simply that the rotation stage is superior to previous ones? The article only stated

"The work is enabled by our recent experimental improvements that speed up tomography so much, that 3D images can be obtained in a movie-like mode with ≥ 25 tomograms per second" and "The new, self-developed high-speed rotation stage", which is a bit vague.

Abstract

In the first sentence there is an ambiguity in the English grammar, the term 'emerging metallic foams' gives the impression that the foams are being extruded or expelled from an opening. I believe the authors mean to stress that metallic foams are 'emerging' as a potentially useful form of material. The opening sentence would be improved by just removing the word 'emerging'.

Tomoscopy

I'm afraid I don't agree with creating a new terminology only to create some arbitrary division of time-scales. For example, if 12500 tomograms are obtained in 12500 seconds (3 1/2 hours at 1hz) the type of data and analysis would be similar (for a process having that time scale) Other authors use '4-d imaging' and I must say I don't like that either, I prefer 'Time-resolved tomography', but if any new term is used I believe it should refer to the extension in time regardless of the absolute speed, so I believe, at least, the notion that one name should be used for greater than 25tps and another for less than 25tps should be eliminated.

Reviewer #2 (Remarks to the Author):

The work presents a novel operando 3D characterization of foaming metal by X-ray tomoscopy. The direct visualization of the process reveals great information that were not accessible before. The comparison of the structure foamed under normal gravity and micro-gravity is also scientifically interesting. The work, if published, will likely lead to great interests from the readers in this and other relevant fields. Some questions/comments for the authors to consider:

1. Can the authors comment on, would the rapid rotation in tomoscopy/tomography affect the foaming process?

2. Overall, the reviewer would expect that more quantitative discussions/analysis regarding the kinetics and fundamental mechanisms on the foaming could be conducted, given the rich

information and unprecedented time resolution provided by the method. Perhaps this can be done in follow-up studies/analysis on these datasets to provide a more in-depth modeling, beyond morphological description. Some of the potential future work on this may be discussed.

3. There have been many high-speed tomography experiments conducted at TOMCAT in the past. Has the time resolution been significantly improved in this work? Or has the same time resolution been demonstrated before so this work is more on the application to the foaming metal? The reviewer is trying understand this point because the manuscript put a great emphasis on the development of the tomography.

Minor comments:

1. In the 2nd paragraph, the authors refer metal foam as a still emerging material – the authors shall reconsider this. Metal foams have well established literatures in a wide variety of topics, and dedicated publications and conferences; it would be fair to say that it remains as an active research field with evolving topics and new categories of materials/processing methods.

2. On page 2 of the manuscript, the author cited 4.9 micron voxel size as 'high spatial resolution'. It should be noted that voxel size is different from the spatial resolution. The authors are encouraged to clarify this.

Reviewer #3 (Remarks to the Author):

The authors presented x-ray imaging experimental results on metallic foam coalescence behaviors. They proposed a new tomography reconstruction technique to improve the measurements' temporal resolution. However, the reviewer found the current work is quite superficial. The new results are not major breakthrough upon what have been reported in literatures. The reviewer recommends publishing the work on a more specialized journal with more focused audience. Below are reviewer's concerns.

1. Aqueous foam systems are mostly studied without tomographic imaging techniques. For instance, avalanche coalescence rupture behaviors in aqueous foam systems were studied with sound detection and high-speed optical imaging. In metallic foam system, x-ray radiology was also applied to study rupture behaviors. If cascade rupture behavior is normal in metallic foam system, why has it never been found in radiographic imaging experiments? Is the 'cascade rupture' behavior found with 'tomography' representative in metallic foam system?

2. The microgravity experimental results and conclusions are basically repeating the results reported in ref. 12 by some authors of the current work.

3. The conclusion of 'the deleterious growth of very large bubbles is mainly caused by the action of the blowing agent and to less extent only by gravity- driven drainage' is also repeating the conclusion in ref. 12.

Dear reviewers,

Thank you for your reviews. We have carried out a number of changes in the paper and the supplementary material. We have assigned a colour to each of the reviewers to make it easier to track the changes in the text, which are written in the colour of the respective reviewer.

We hope that we can convince you that our paper is suitable for publication in Nature Communications now.

Yours sincerely,

F. García-Moreno

Reviewer #1 (Remarks to the Author):

Summary

The authors have demonstrated capability to obtain 3-d (tomographic) X-ray images at an order of magnitude higher rate than reported previously. They have applied this technique to observe the structure changes in a metallic foam produced by heating an alloy containing outgassing agents. The technique being performed at a synchrotron (where the X-ray beam is necessarily fixed) requires that the sample be rotated at 100rpm. The influence of the inertial forces pertaining on the sample have been compared by contrasting with radiography results obtained in microgravity. They have shown dramatic new capability in X-ray imaging, and applied it to a problem of both technological importance and fundamental scientific interest. The paper should be published and I think it is suitable for Nature Communications.

One thing that was not clear to me, is what is the main reason or key technology allowing the greater speed, is there greater flux at Tomcat than previously? or simply that the rotation stage is superior to previous ones? The article only stated "The work is enabled by our recent experimental improvements that speed up tomography so much, that 3D images can be obtained in a movie-like mode with ≥ 25 tomograms per second" and "The new, self-developed high-speed rotation stage", which is a bit vague.

The flux at the synchrotron has not changed. The rotation stage was an important element. Another key point was the ability to stream the data directly from the camera chip to the computer system. Finally, automated image analysis had to be developed. With all this together, the tomography acquisition rate could be increased by a factor of 10 and the previous restriction of only short experiments overcome. Now we can acquire 40,000 tomograms and more in a series.
→ New passage in the main text, p. 2, and description of rotation stage in supplementary material, p. 2.

Abstract

In the first sentence there is an ambiguity in the English grammar, the term 'emerging metallic foams' gives the impression that the foams are being extruded or expelled from an opening. I believe the authors mean to stress that metallic foams are 'emerging' as a potentially useful form of material. The opening sentence would be improved by just removing the word 'emerging'.

The intention was to emphasize that metallic foams are evolving in the liquid state, not that foams

are an emerging material.

→ We have replaced the ambiguous word “emerging” by “evolving”.

I'm afraid I don't agree with creating a new terminology only to create some arbitrary division of time-scales. For example, if 12500 tomograms are obtained in 12500 seconds (3 1/2 hours at 1hz) the type of data and analysis would be similar (for a process having that time scale) Other authors use '4-d imaging' and I must say I don't like that either, I prefer 'Time-resolved tomography', but if any new term is used I believe it should refer to the extension in time regardless of the absolute speed, so I believe, at least, the notion that one name should be used for greater than 25tps and another for less than 25tps should be eliminated.

Our intention was to move away from terms like “fast tomography”, “ultrafast tomography” etc. and to create a term that will be also valid in 20 years after acquisition rates have increased again and what is fast today is then slow.

→ We understand the concern that 25 tps is an arbitrary limit and have decided to drop this part of the definition. We now define in the main text, p. 2 (and shortly in the abstract), that tomography is:

- time-resolved 3D tomography,
- that is applied to an evolving system to clarify its dynamics,
- that is “continuous”, i.e. not just the state before and after a change are captured but we are monitoring various stages during evolution of the system (without specifying how many)

Now there is an analogy between 2D and 3D:

(2D) radiography = static → radioscopy = dynamic

(3D) tomography = static → tomography = dynamic

Reviewer #2 (Remarks to the Author):

The work presents a novel operando 3D characterization of foaming metal by X-ray tomography. The direct visualization of the process reveals great information that were not accessible before. The comparison of the structure foamed under normal gravity and micro-gravity is also scientifically interesting. The work, if published, will likely lead to great interests from the readers in this and other relevant fields. Some questions/comments for the authors to consider:

1. Can the authors comment on, would the rapid rotation in tomography/tomography affect the foaming process?

This is an important point since radial accelerations can be high. We are dealing with early stages of foaming in the 208 tps studies, which implies that the foam is still rather incompressible. Moreover, the alloy under consideration is known to be quite stable against drainage effects. This is why we were not surprised (and relieved) that drainage does not lead to visual segregation of bubbles and liquid. In order to support this qualitative argument we have calculated a radial density profile of a sample, which was rotated slowly (1 tps), at medium speed (50 tps) and fast (208 tps).

→ the supplementary information now contains a description of these density profiles on p. 2 and in Fig. S6.

2. Overall, the reviewer would expect that more quantitative discussions/analysis regarding the kinetics and fundamental mechanisms on the foaming could be conducted, given the rich information and unprecedented time resolution provided by the method. Perhaps this can be done in follow-up studies/analysis on these datasets to provide a more in-depth modeling, beyond morphological description. Some of the potential future work on this may be discussed.

We are discussing fundamental phenomena such as nucleation already but for sure we are planning to extract more data from the tomography series in the future. The limitations at the moment are the algorithms needed to process 40,000 tomograms automatically and in a reliable way. State-of-the-art are bubble size, shape and number density calculations on a bulk of tomograms. Moreover, analysis of individual local features such as in Figs. 3, 4 and 6. We still have to develop analyses in cases of weak contrast and correlation analyses (i.e. between blowing agent particles and bubbles) that include spatial and temporal correlations. This will be part of future work, as well as more work on a greater variety of alloys, temperature profiles etc. that could be included in follow-up studies as suggested.

3. There have been many high-speed tomography experiments conducted at TOMCAT in the past. Has the time resolution been significantly improved in this work? Or has the same time resolution been demonstrated before so this work is more on the application to the foaming metal? The reviewer is trying understand this point because the manuscript put a great emphasis on the development of the tomography.

The highest repetition rate at PSI so far has been 20 tps (see literature overview in supplement). Therefore we demonstrate a 10 times higher time resolution now (first time showing this). The development of a suitable rotation stage was an important element. Moreover, we do this continuously over a long time (60 s), which previously was not possible due to the limited memory of the camera chip and the lack of an on-the-fly processing of data.

→ This is now stated explicitly on p. 2.

Minor comments:

1. In the 2nd paragraph, the authors refer metal foam as a still emerging material – the authors shall reconsider this. Metal foams have well established literatures in a wide variety of topics, and dedicated publications and conferences; it would be fair to say that it remains as an active research field with evolving topics and new categories of materials/processing methods.

Correct, metal foams have been known for many years but still their use is quite restricted. This is what we wish to express and after a slight change we hope will be understood in this way.

→ An extra reference (Ref. 2) on p. 1 helps to understand how far the history of metal foams dates back.

2. On page 2 of the manuscript, the author cited 4.9 micron voxel size as ‘high spatial resolution’. It should be noted that voxel size is different from the spatial resolution. The authors are encouraged to clarify this.

This is correct and as we admit was a bit sloppy. The true resolution has unfortunately not been measured this time, but was determined previously with a similar configuration to be approximately 12 μm .

→ rephrasing on p. 2.

Reviewer #3 (Remarks to the Author):

The authors presented x-ray imaging experimental results on metallic foam coalescence behaviors. They proposed a new tomography reconstruction technique to improve the measurements' temporal resolution. However, the reviewer found the current work is quite superficial. The new results are not major breakthrough upon what have been reported in literatures. The reviewer recommends publishing the work on a more specialized journal with more focused audience. Below are reviewer's concerns.

We deliberately chose a journal for a general audience (Nat Comm) because the new possibilities of tomoscopy at unprecedented rates (now 208 tps), and already aiming at even faster rates, will stimulate studies in other areas of materials science, mechanical engineering, and applied physics.

In the following we will try to convince the referee that also the specific application of tomoscopy to metal foam yields some completely new insights.

1. Aqueous foam systems are mostly studied without tomographic imaging techniques. For instance, avalanche coalescence rupture behaviors in aqueous foam systems were studied with sound detection and high-speed optical imaging. In metallic foam system, x-ray radiology was also applied to study rupture behaviors. If cascade rupture behavior is normal in metallic foam system, why has it never been found in radiographic imaging experiments?

Aqueous foams have also been investigated by tomography in the past years by various groups [1-10] including our own group [11-15]. Although water is transparent it is impossible to unravel the structure of deeper layers due to strong light scattering. Using X-ray tomography (slower as in this paper however) we showed e.g. that ordered packings occur after ageing initially disordered foams for a few days [14].

Rupture cascades in metal foams actually have been observed before in radioscopic imaging studies [16], added now as new Ref (31) to the manuscript. However, showing that various rupture events are correlated is not always straight-forward as along the viewing direction features are superimposed. Therefore, tomoscopy as applied in this paper provides better evidence.

→ We have added Ref. [16] as new Ref (31) to the manuscript and a sentence on p. 4 to explain this fact.

Is the 'cascade rupture' behavior found with 'tomoscopy' representative in metallic foam system?

We have found many cascades by tomoscopy in the samples investigated (two already shown in Figs. 4 and 6). We have extracted many other events. Here we show 4 examples:

We think it would overload the paper if we included too many such examples and that 2 are enough (Fig. 4 and 6).

Radioscopy on larger samples [16] showed many more events in an AlSiCu alloy (with the caveat that some events might be accidentally aligned features and not real neighbours in 3D).

2. The microgravity experimental results and conclusions are basically repeating the results reported in ref. 12 by some authors of the current work.
3. The conclusion of ‘the deleterious growth of very large bubbles is mainly caused by the action of the blowing agent and to less extent only by gravity- driven drainage’ is also repeating the conclusion in ref. 12.

In former Ref. (12) of the manuscript we concluded that coalescence cannot be the consequence of the removal of liquid from a foam driven by gravity. This was based on the observation of foams produced under gravity and without that showed roughly the same level of coalescence. However, the conclusion was based on an indirect argument (no difference between different gravity levels) not on a direct observation of the actual mechanism. This is what we provide now by visualizing the expanding and strongly inflated bubbles around the blowing agent particles in 3D. The results in Fig. 5 are new and have not been shown in former Ref. (12) that - being a review article on the application of X-ray radioscopy – was very brief.

Moreover, we would like to emphasise that this is just one result of the paper that we have emphasised in the abstract in the usual “Here we show ...” statement. We put the emphasis on this because it is a finding that can be translated into technological improvements directly.

→ we have squeezed in some words into the abstract to make clear that this is just one of the results of this paper.

Other important results include:

- Volume and bubble number evolve in two stages, which we now know are connected to different gas sources. (Fig. 2)
- Melting of the AlMg constituent is the trigger for first expansion (Fig. 2)
- Liquid actually retracts from films prior to rupture and it was possible to quantify the kinetics of retraction (Fig. 3)
- Rupture cascades have been directly shown in 3D images now (Fig. 4)
- Coalescence is caused by 3 mechanisms (drainage coalescence, global growth coalescence and local coalescence), and we can say something about the relative importance of the mechanisms (Figs. 2e, 5). This is a completely new finding
- The influence of the oxide layer on bubble shrinkage has been shown by following bubble shrinkage directly (Fig. S5)

→ the conclusions have been slightly modified to emphasise the link between the coalescence induced by the blowing agent and technology.

In summary, various open questions on the behaviour of liquid metal foams have been answered and we would not call this “superficial”.

References:

- [1] Lambert J, Cantat I, Delannay R, Renault A, Graner F, Glazier JA, et al. Extraction of relevant physical parameters from 3D images of foams obtained by X-ray tomography. *Colloids Surf, A* 2005;263:295-302.
- [2] Lambert J, Cantat I, Delannay R, Mokso R, Cloetens P, Glazier JA, et al. Experimental Growth Law for Bubbles in a Moderately “Wet” 3D Liquid Foam. *Phys Rev Lett* 2007;99:058304.
- [3] Lambert J, Mokso R, Cantat I, Cloetens P, Glazier JA, Graner F, et al. Coarsening Foams Robustly Reach a Self-Similar Growth Regime. *Phys Rev Lett* 2010;104:248304.
- [4] Mader K, Mokso R, Raufaste C, Dollet B, Santucci S, Lambert J, et al. Quantitative 3D characterization of cellular materials: Segmentation and morphology of foam. *Colloids Surf, A* 2012;415:230-8.
- [5] Davies IT, Cox SJ, Lambert J. Reconstruction of tomographic images of dry aqueous foams. *Colloids Surf, A* 2013;438:33-40.
- [6] Turbin-Orger A, Babin P, Boller E, Chaunier L, Chiron H, Della Valle G, et al. Growth and setting of gas bubbles in a viscoelastic matrix imaged by X-ray microtomography: the evolution of cellular structures in fermenting wheat flour dough. *Soft Matter* 2015;11:3373-84.
- [7] Raufaste C, Dollet B, Mader K, Santucci S, Mokso R. Three-dimensional foam flow resolved by fast X-ray tomographic microscopy. *EPL (Europhysics Letters)* 2015;111:38004.
- [8] Thomas GL, Belmonte JM, Graner F, Glazier JA, de Almeida RMC. 3D simulations of wet foam coarsening evidence a self similar growth regime. *Colloids Surf, A* 2015;473:109-14.
- [9] Eggert A, Müller M, Nachtrab F, Dombrowski J, Rack A, Zabler S. High-speed in-situ tomography of liquid protein foams. *Int J Mater Res* 2014;105:632-9.
- [10] Dittmann J, Eggert A, Lambertus M, Dombrowski J, Rack A, Zabler S. Finding robust descriptive features for the characterization of the coarsening dynamics of three dimensional whey protein foams. *J Colloid Interface Sci* 2016;467:148-57.

- [11] Stocco A, Garcia-Moreno F, Manke I, Banhart J, Langevin D. Particle-stabilised foams: structure and aging. *Soft Matter* 2011;7:631-7.
- [12] Meagher AJ, Mukherjee M, Weaire D, Hutzler S, Banhart J, Garcia-Moreno F. Analysis of the internal structure of monodisperse liquid foams by X-ray tomography. *Soft Matter* 2011;7:9881-5.
- [13] Meagher AJ, García-Moreno F, Banhart J, Mughal A, Hutzler S. An experimental study of columnar crystals using monodisperse microbubbles. *Colloids Surf, A* 2015;473:55-9.
- [14] Meagher AJ, Whyte D, Banhart J, Hutzler S, Weaire D, Garcia-Moreno F. Slow crystallisation of a monodisperse foam stabilised against coarsening. *Soft Matter* 2015;11:4710-6.
- [15] García-Moreno F, Kamm PH, Neu T, Heim K, Rack A, Banhart J. In situ X-ray tomography of aqueous foams: Analysis of columnar foam generation. *Colloids Surf, A* 2017;534:78-84.
- [16] Paepflow M, García-Moreno F, Meagher A, Rack A, Banhart J. Coalescence Avalanches in Liquid Aluminum Foams. *Metals* 2017;7:298.

Reviewers' comments:

Reviewer #2 (Remarks to the Author):

The authors addressed most of the questions from the reviewer.

In particular, the efforts to include the additional figure, Fig. S6, is appreciated and the point is well taken. Some details shall be further clarified:

1. It is unclear how the calculation was done, e.g. how was the 'normalized' step conducted in the 'normalised integrated intensities' along the radius? What step size along the radius was calculated? More details should be provided.

2. The circles drawn in the insets seem to be not centered around the center of the reconstructed sample (rotation axis)? Any particular reason for this?

Additional questions for the reviewer/reader's information:

3. How sensitive this calculation would be? Namely, what density differences would the authors be able to detect, if there are differences?

4. The early/late stage comment is very interesting – it'd be valuable to quantify the on-set of such transition – do the authors have a later stage of the data that does demonstrate the density gradient along the radius direction where such phenomena would be expected?

The reviewer still found the claim of 'at small voxel sizes of 4.9 microns' in that particular contents was unnecessary and confusing. The authors' point was that with fast time resolution, they still obtained good spatial resolution. While reporting voxel size is important and should be done, it does not offer spatial resolution information. 'Small' is a relative, subjective term. More importantly, one can collect the tomography with 'small' voxel size but yet if all signal/data is smearing due to poor design of the instrument, it does not actually provide the optical resolution. The authors have demonstrated such capability - while the resolution was not measured, the authors have much data to get an estimate of the approximate resolution – e.g. features/gaps around x-x micron range can be well resolved by x% contrast, etc. If the technical advancement is one of the keys for this paper, the reviewer would urge the author to avoid providing such confusing number to the community – for one, this does not provide information, and for the non-experts, it is misleading.

The authors pointed out some exciting future directions but yet none was incorporated into the revised manuscript. Especially, the challenging task of analyzing 40,000 data is interesting and important for the community – some aspect of automated analysis may be suggested.

With another look of the manuscript, the reviewer has other suggestions:

1. Scale bars for 3D images: the reviewer found the way of making the 3D volume rendering figures confusing, e.g. in Figure 4 – there are bounding boxes, but yet the scale bar is marked in 2D. A better way to label this is to label the bounding box size in all three axes.

2. Gray scale map color scale: The color scale is missing in all reconstructed pseudo cross-sections – just wondering if the reconstructed values directly represent the X-ray attenuation here? Is it quantitative? If so, it'd be good to include such information.

Reviewer #3 (Remarks to the Author):

No remarks to the authors.

Reviewer #4 (Remarks to the Author):

As an expert on the coherent x-ray imaging techniques that have been employed as the core enabling optics of this paper, I can very confidently assert that the author responses to Reviewer #1 has been most satisfactorily addressed in the revised manuscript. This extremely impressive paper should certainly now be published, in my very strong opinion.

Even though my brief was merely to assert whether or not the report of Reviewer #1 had been satisfactorily taken into account in the revised manuscript, I have also independently studied both the paper and the referee reports, and my independent judgement is that this paper is an absolute tour-de-force (over 200 tomographies per second is obviously an extremely important and indeed astonishing breakthrough, and with an excellent voxel resolution of 4.9 microns). Again: This result, as an enabling technology for x-ray imaging, is indeed a landmark result, as is the particular study of metallic foams to which it has been applied. Applications of the technology developed in this paper will surely reach far beyond the particular application, explored in this paper, in the future. In my considered view, the authors have now done more than enough in response to the referee comments, for the paper to be accepted and published without any further revisions being necessary.

One small typographic change: `processing reconstruction algorithmen` should be `processing reconstruction algorithm`, near the end of the main text.

David Paganin

Reviewers' comments:

Reviewer #2 (Remarks to the Author):

The authors addressed most of the questions from the reviewer.

In particular, the efforts to include the additional figure, Fig. S6, is appreciated and the point is well taken. Some details shall be further clarified:

1. It is unclear how the calculation was done, e.g. how was the 'normalized' step conducted in the 'normalised integrated intensities' along the radius? What step size along the radius was calculated? More details should be provided.

The intensities of the inverted grey images were averaged over the number of slices, whereupon they were represented over the radius normalized to the circumference. In the diagram, these were again normalized to the arithmetic mean of the respective intensities of the (nearly dense) original sample precursor with a porosity of 1 %. Some text was added to the manuscript.

2. The circles drawn in the insets seem to be not centered around the center of the reconstructed sample (rotation axis)? Any particular reason for this?

The circles share the centre with the rotation axis (centre of the reconstructed section), which does not correspond exactly with the centre of the sample. A displacement of about 90 μm is due to the manual mounting of the crucible on the rotation axis.

Additional questions for the reviewer/reader's information:

3. How sensitive this calculation would be? Namely, what density differences would the authors be able to detect, if there are differences?

Variations of up to ~10% in density inside a single metal foam are normal. Intensity differences of about ~1% are detectable. There is an effect at higher speeds, which was recently confirmed by other, faster experiments, not part of this work. A further problem for this material is the size of the differently absorbing, contiguous volumes (material and bubbles) and the small sample sizes used, which allows little statistics. Some text was added to the manuscript.

4. The early/late stage comment is very interesting – it'd be valuable to quantify the on-set of such transition – do the authors have a later stage of the data that does demonstrate the density gradient along the radius direction where such phenomena would be expected?

The 1% and 40% samples represent the early (almost solid) and late (almost liquid) stages respectively. Fig S4 shows the porosity evolution. The comparison of the intensities up to 200 tps do not show any radial changes or density gradients apart from the natural liquid foam scattering. A transition to a radial density gradient appears at higher g levels and later stages. These analyses are currently the subject of our research and still under investigation.

The reviewer still found the claim of 'at small voxel sizes of 4.9 microns' in that particular contents was unnecessary and confusing. The authors' point was that with fast time resolution, they still obtained good spatial resolution. While reporting voxel size is important and should be done, it does not offer spatial resolution information. 'Small' is a relative, subjective term. More importantly, one can collect the tomography with 'small' voxel size but yet if all signal/data is smearing due to poor design of the instrument, it does not actually provide the optical resolution. The authors have demonstrated such capability - while the resolution was not measured, the authors have much data to get an estimate of the approximate resolution – e.g. features/gaps around x-x micron range can be well resolved by x% contrast, etc. If the technical advancement is one of the keys for this paper, the reviewer would urge the author to avoid providing such confusing number to the community – for one, this does not provide information, and for the non-experts, it is misleading.

This is true, a "small" voxel size does not represent by itself alone a good resolution. The resolution also depends e.g. on the exposure time, thickness of the scintillator, number of projections, distance to the center of rotation, as the distance between projection planes is larger, etc.. Therefore it is also impossible to give a single spatial resolution for a whole tomogram, as this is dependent on the

distance to the center of rotation, i.e. on the radius. But the pixel or voxel size are a good hint for the reader assuming the instrument is properly design, what to our knowledge is the case at the Tomcat beamline. Most of the publications concerning tomography provide only the pixel size, even not the voxel size, which may difer (see e.g. Fig. 1 of [1]). Newertheless, following the suggestion of the reviewer, we selected some features (gap between precursor sample and crucible) and measured that at extrem conditions to be over 3 pixels ($14.7 \mu\text{m}$), where we have 6% contrast, allowing to distinguish the gap. Although this gap is quite far away from the rotation center, not as sharp as e.g. a resolution pattern and the tomograms were acquired at 200 tps, we could estimate the spatial resolution to be $\sim 15 \mu\text{m}$ or better. This was added to the text in the manuscript and supplementary material. The present publication concentrates more on the scientific result of the experiments. A more methodical publication is in progress.

The authors pointed out some exciting future directions but yet none was incorporated into the revised manuscript. Especially, the challenging task of analyzing 40,000 data is interesting and important for the community – some aspect of automated analysis may be suggested.

Basically we used a MATLAB script for batch analysis concerning binarisation, segmentation, pore counting, equivalent diameter, etc. for selected series as mentioned in the supplementary material already. As mentioned in the Code/Data Availability the experimental data and the workflow using MATLAB to quantify in a batch modus are available upon reasonable request. As previously stated, a more methodical publication is in progress, where the method and a more sophisticated workflow, will be explained more in detail.

With another look of the manuscript, the reviewer has other suggestions:

1. Scale bars for 3D images: the reviewer found the say of making the 3D volume rendering figures confusing, e.g. in Figure 4 – there are bounding boxes, but yet the scale bar is marked in 2D. A better way to label this is to label the bounding box size in all three axes.

Since this is an orthographic axonometry, three-dimensional objects are projected vertically and parallel onto the image plane. In the viewing direction, it is therefore easier for the observer to estimate the size of objects, especially round pores - quantitative measurements on planes that are not orthogonal to the viewing direction are therefore not possible. For this purpose the size of the bounding box is specified in Figure 4 in the caption. However, we tried also to place a grid with defined distances (in the shown case $50 \mu\text{m}$) on the faces. But from our point of view, this does not create any added value for the reader.

2. Gray scale map color scale: The color scale is missing in all reconstructed pseudo cross-sections – just wondering if the reconstructed values directly represent the X-ray attenuation here? Is it quantitative? If so, it'd be good to include such information.

We have not included color scales, since the gray values do not represent a quantitative property. In the supplementary material we indicated that we used a propagation-based phase contrast algorithm.

The use of the single-distance propagation-based phase-retrieval method increases the contrast at edges and surfaces in the sample.

References:

[1] Villanova J, Daudin R, Lhuissier P, Jauffrès D, Lou S, Martin CL, et al. Fast in situ 3D nanoimaging: a new tool for dynamic characterization in materials science. *Mater Today* 2017;20:354-9.

REVIEWERS' COMMENTS:

Reviewer #2 (Remarks to the Author):

The authors have addressed all my questions and comments. The reviewer has no further comments and thinks highly of this work.